# Giant intrinsic photovoltaic effect in one-dimensional van der Waals grain boundaries

Yongheng Zhou[1,7], Xin Zhou [2,3,7], Xiang-Long Yu [4,5] ✉, Zihan Liang[1], Xiaoxu Zhao [3], Taihong Wang [1], Jinshui Miao [6] ✉ & Xiaolong Chen [1] ✉

The photovoltaic effect lies at the heart of eco-friendly energy harvesting. However, the conversion efficiency of traditional photovoltaic effect utilizing the built-in electric effect in p-n junctions is restricted by the Shockley-Queisser limit. Alternatively, intrinsic/bulk photovoltaic effect (IPVE/BPVE), a second-order nonlinear optoelectronic effect arising from the broken inversion symmetry of crystalline structure, can overcome this theoretical limit. Here, we uncover giant and robust IPVE in one-dimensional (1D) van der Waals (vdW) grain boundaries (GBs) in a layered semiconductor, $ReS_2$. The IPVE-induced photocurrent densities in vdW GBs are among the highest reported values compared with all kinds of material platforms. Furthermore, the IPVE-induced photocurrent is gate-tunable with a polarization-independent component along the GBs, which is preferred for energy harvesting. The observed IPVE in vdW GBs demonstrates a promising mechanism for emerging optoelectronics applications.

In a non-centrosymmetric material, light-matter interactions can generate a finite DC photocurrent under homogeneous illumination in absence of external bias and spatial inhomogeneity. This photovoltaic effect, governed by the intrinsic symmetry properties of materials, is referred to the intrinsic photovoltaic effect (IPVE) or bulk photovoltaic effect (BPVE)[1–4]. Hence, the unique physics of IPVE offers an effective approach to surpass the Shockley-Queisser limit in traditional photovoltaic devices[1–9], which attracts growing attention recently. Initial studies on IPVE mainly focused on ferroelectric insulators, such as $LiNbO_3$[2], $BiFeO_3$[10] and $BaTiO_3$[8,11]. Later, researchers found that reducing bandgap size and lowering dimensionality could further enhance the efficiency of IPVE[5–7,12–17]. For example, the IPVE photocurrents observed in narrow bandgap semiconductors (including one-dimensional/1D $WS_2$ nanotubes[5]) and Weyl semimetals with broken inversion symmetry are orders of larger than those in wide bandgap ferroelectric insulators[6,7,12–15]. On the other hand, van der Waals (vdW) layered materials meet all merits for IPVE investigations due to its low

dimensionality, tunable bandgap, flexibility, easy manipulation, and rich species[15–26]. For example, strain-gradient-engineered $MoS_2$ shows a strong IPVE with photocurrent density over $10^2\,A\,cm^{-2}$, which is comparable to that in 1D $WS_2$ nanotube[5,20]; The external quantum efficiency of $3R\text{-}MoS_2$ with spontaneous out-of-plane polarization shows the highest reported value of 16%[22]; IPVE observed in the in-plane strained $3R\text{-}MoS_2$ is over two orders of magnitude higher than the unstrained one[20]; The non-centrosymmetric nano-antennas in centrosymmetric graphene can result in artificial IPVE[23–25]; Moiré-pattern in twisted bilayer graphene and $WSe_2$/BP interface can lead to the emergence of spontaneous IPVE[21,26]; Low-dimensional vdW structures such as quasi-1D edges of Weyl semimetal $WTe_2$ can generate strong IPVE-induced photocurrents, attributing to the strong symmetry breaking and low dimensionality of edges[15]; Robust IPVE-induced photocurrents are observed in topological insulator monolayer $WTe_2$[16]. Here, we introduce an alternative low-dimensional system, one-dimensional grain boundary (GB) with non-centrosymmetric

[1]Department of Electrical and Electronic Engineering, Southern University of Science and Technology, 1088 Xueyuan Avenue, Shenzhen 518055, China. [2]Department of Materials Science and Engineering, National University of Singapore, Singapore 117575, Singapore. [3]School of Materials Science and Engineering, Peking University, Beijing 100871, China. [4]Shenzhen Institute for Quantum Science and Engineering, Southern University of Science and Technology, 1088 Xueyuan Avenue, Shenzhen 518055, China. [5]International Quantum Academy, Shenzhen 518048, China. [6]State Key Laboratory of Infrared Physics, Shanghai Institute of Technical Physics, Chinese Academy of Sciences, Shanghai 200083, China. [7]These authors contributed equally: Yongheng Zhou, Xin Zhou. ✉e-mail: yuxl@sustech.edu.cn; jsmiao@mail.sitp.ac.cn; chenxl@sustech.edu.cn

crystalline structure, for IPVE investigations. Distinct from previous IPVE systems, GBs widely exist in all kinds of materials. For example, GBs have been uncovered in various vdW layered materials regardless of their crystalline symmetry, including graphene[27], MoS$_2$[28], ReS$_2$[29–33], and MoSe$_2$[34].

1 T'-ReS$_2$ GBs are ideal for IPVE investigations due to following reasons. (1) Anisotropic optical properties of ReS$_2$ allow to identify positions of GBs and subdomains simply using polarization-resolved optical microscopy; (2) GBs in ReS$_2$ have well-defined structures free of dangling bonds. In this work, we uncover strong and robust IPVE in 1D vdW GBs in ReS$_2$. Symmetry analysis and experimental results demonstrate that inversion symmetry is broken near GBs, which results in a DC photocurrent that propagates along GBs without any voltage bias. We demonstrate that this IPVE-induced photocurrent is gate tunable and possesses a pronounced polarization-independent component. Furthermore, the IPVE-induced photocurrent densities in 1D ReS$_2$ GBs are among the highest values compared with reported material systems.

## Results

### Characterization of 1D vdW GBs in ReS$_2$

Bulk 1 T'-ReS$_2$ is a vdW semiconductor with a centrosymmetric crystalline structure under the inversion symmetric space group of $P\bar{1}$[35–38], as demonstrated by the scanning transmission electron microscope (STEM) image in Fig. 1a. Thus, IPVE is not allowed in thin-film ReS$_2$. In addition, we find an abundance of 1D GBs in ReS$_2$. The in-plane orientations, signified by the direction of Re chains, of the two neighboring subdomains form a 120° angle (see Fig. 1b and Supplementary Fig. 1), which aligns with prior STEM research findings[30,33]. Figure 1c shows a top view schematic of ReS$_2$ crystalline structure with GBs. Here, we use A and B to represent two adjacent subdomains. BA and AB GBs are denoted by "↑" and "↓" arrows, respectively. Using polarization-resolved optical microscopy (see Fig. 1d, e and

Supplementary Fig. 2), we can clearly identify the positions of GBs in ReS$_2$ flakes due to the anisotropic optical reflection and different Re-chain directions of subdomains. As shown in Fig. 1e, the ReS$_2$ flake is separated by multiple GBs (along the $y$-direction) and forms multi-domain structures. Angle-resolved polarized Raman spectroscopy was further performed to identify the crystalline orientations of ReS$_2$ subdomains (see Fig. 1f). The intensity of $A_{g2}$ mode (212 cm$^{-1}$) is maximum when the light polarization direction is parallel to the Re-chain direction of ReS$_2$[30,32,36]. Raman result indicates that there is ~117° difference between Re-chain directions of two adjacent subdomains, which is very close to the angle ~120° observed in STEM. The ~3° deviation of Raman characterization is within the permissible range of our instruments.

### IPVE theory in 1D vdW GBs

The only symmetry present near GB regions is the two-fold rotation along the $y$-directional axis. Expanded analysis in Supplementary Note 1 and Supplementary Fig. 3 suggests that the GB region is characterized by a point group of $C_2$ and associated with a broken inversion symmetry, resulting in nonzero second-order nonlinear light-matter-interaction tensors $\sigma_{ljk}^{(2)}(w, \vec{q})$, where $w$ is the angular frequency of incident light, $\vec{q}$ is the wave vector and $l/j/k$ represents $x$-, $y$-, or $z$-directions. Under linearly polarized light, a finite DC photocurrent density along $l$-direction can be generated (only consider the $\vec{q}$-independent term)[20,39]

$$J_l^{LBPVE} = \frac{1}{2}\sum_{j,k}\chi_{ljk}(E_jE_k^* + E_kE_j^*) \quad (1)$$

where $\chi_{ljk} = \sigma_{ljk}^{(2)}(w,0)$. Since $J_l^{LBPVE}$ only depends on the intrinsic physical properties of materials, this effect is called IPVE or BPVE. Here, we mainly focus on IPVE-induced photocurrent along GB ($y$-direction) $J_y^{LBPVE}$. For incident light normal to the two-dimensional plane of ReS$_2$

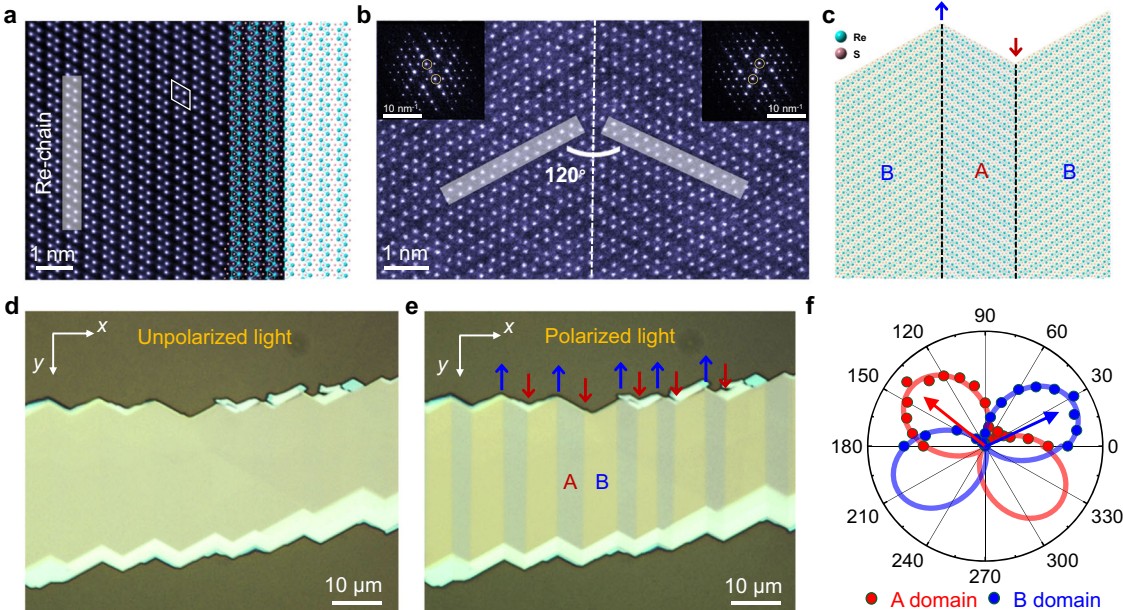

**Fig. 1 | Characterization of 1D van der Waals (vdW) grain boundaries (GBs) in ReS2. a** Atomic-resolution scanning transmission electron microscope (STEM)/high angle annular dark field (HAADF) image of bulk ReS$_2$, accompanied with a lattice structure schematic. The blue and pink spheres in schematic represent the Re and S atoms, respectively. The unit cell of ReS$_2$ is delineated by a white rhombus. Scale bar is 1 nm. **b** The STEM/HAADF image of a ReS$_2$ GB, indicated by a white dashed line. There is 120° between Re-chain directions of two adjacent subdomains. Scale bar is 1 nm. Insets show Fourier transform patterns of two adjacent

subdomains with scale bars of 10 nm$^{-1}$. **c** Top view schematic of lattice structures near GBs. The inversion symmetry is broken near GBs. **d, e** Unpolarized (**d**) and polarized (**e**) optical images of a ReS$_2$ flake with subdomain structures. Two adjacent subdomains are marked by A (red) and B (blue), respectively. BA and AB GBs are denoted by ↑ and ↓ arrows, respectively. Scale bar is 10 μm. **f** Angle-resolved polarized Raman spectroscopy results of $A_{g2}$ mode (212 cm$^{-1}$) in A (red dots) and B (blue dots) subdomains. The Re-chain directions of A and B subdomains are denoted by red and blue arrows, respectively. Solid lines are fitting curves.

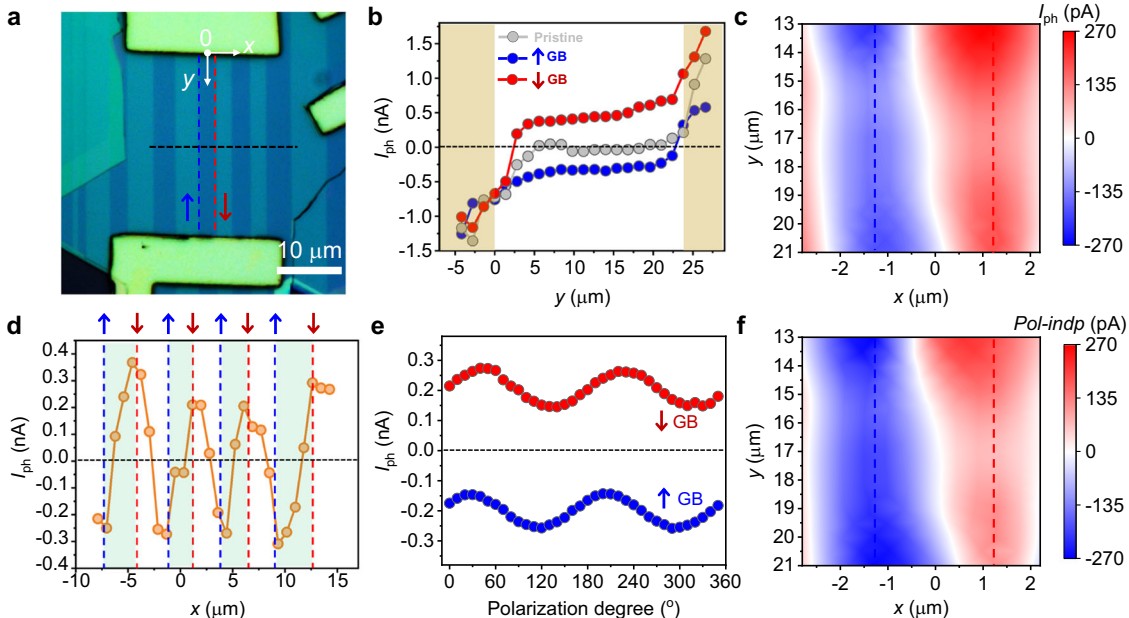

**Fig. 2 | Observation of intrinsic photovoltaic effect (IPVE) in 1D ReS2 GBs.**
**a** Optical image of a photodetector with multiple domains under polarized white light. Two adjacent GBs with reversed orientations are denoted by "↑" and "↓" and marked by blue and red dash lines, respectively. Scale bar is 10 μm. **b** Short-circuit photocurrents ($I_{ph}$) measured along $y$-direction at pristine region (gray dots), ↑ (blue dots) and ↓ (red dots) GBs. Robust $I_{ph}$ are observed at ↑ and ↓ GBs. Electrodes are marked by yellow regions. **c** Scanning photocurrent spectroscopy of total photocurrent $I_{ph}$. Incident laser is polarized along $x$-axis. The ↑ and ↓ GBs are marked by blue and red dash lines, respectively. **d** $I_{ph}$ measured along $x$-direction as indicated by the black dashed line in **a**. Valley and peak features are observed at ↑ and ↓ GBs, respectively. **e** Polarization-resolved photocurrents in middle positions of ↑ and ↓ GBs. The polarization degree is the angle between laser polarization and $y$-axis. **f** Spatial distribution of the polarization-independent term $J_y^{Pol-indp}$. The ↑ and ↓ GBs are marked by blue and red dash lines, respectively.

flake ($E_z = 0$), $J_y^{LBPVE}$ can be written as

$$J_y^{LBPVE} = \chi_{yxx}|E_x|^2 + \chi_{yyy}|E_y|^2 + \chi_{yxy}E_xE_y^* + \chi_{yxy}E_yE_x^* \quad (2)$$

Equation 2 can be further simplified utilizing the rotation symmetry in GB region as shown in Fig. 1b (see Supplementary Note 2 for details). Under rotation symmetry ($x$, $y$, $z \rightarrow -x$, $y$, $-z$), $J_y^{LBPVE}(x,y,z) = J_y^{LBPVE}(-x,y,-z)$ which makes $\chi_{yxy} = 0$. If we write $E = [E_0\sin\theta, E_0\cos\theta, 0]$, where $\theta$ is the angle between $y$-direction and light polarization direction, then we have

$$J_y^{LBPVE} = \frac{E_0^2}{2}(\chi_{yyy} - \chi_{yxx})\cos 2\theta + \frac{E_0^2}{2}(\chi_{yyy} + \chi_{yxx}) = J_y^{Pol-dp} + J_y^{Pol-indp} \quad (3)$$

Here, $J_y^{Pol-dp}$ and $J_y^{Pol-indp}$ are the polarization-dependent and polarization-independent terms, respectively. Furthermore, IPVE-induced photocurrents along two adjacent ↑ and ↓ GBs should have opposite directions restricted by their reversed orientations.

## Experimental observation of IPVE in 1D vdW GBs

To study the IPVE in ReS2 GBs, photodetectors with channel parallel to GBs are fabricated. Figure 2a shows the optical image of a device under polarized white light (the angle between light polarization and $y$-axis is ~30°). The optical images at other polarized angles are shown in Supplementary Fig. 2. The thickness of ReS2 flake is ~180 nm determined by atomic force microscope (AFM) (see Supplementary Fig. 4). The ↑ and ↓ GBs are denoted by blue and red dash lines, respectively. The detailed fabrication process can be found in the Method section. The device shows linear current-voltage ($I_{ds}$-$V_{ds}$) characteristic, indicating a good Ohmic contact between ReS2 and metal electrodes (see Supplementary Fig. 5). A linearly polarized 532 nm laser with a diameter around 3 μm was focused on the channel and short-circuit photocurrents ($I_{ph}$) were collected (the angle between laser polarization and

$y$-axis is ~30°). As shown in Fig. 2b, $I_{ph}(y)$ at pristine region ($x = 0$ μm) shows ordinary shape with vanishing value in the middle of channel. The finite photocurrents near electrodes can be attributed to extrinsic photovoltaic effect, such as built-in Schottky junction between ReS2 and electrodes and photo-thermoelectric effect. This indicates that the pristine region of ReS2 does not support IPVE due to the preservation of inversion symmetry. This observation is further confirmed in a device based on ReS2 without GBs (see Supplementary Fig. 6). On the other hand, we observed very robust photocurrents in the middle regions of GBs with negative values at ↑ GB (along blue dashed line in Fig. 2a) and positive values at ↓ GB (along red dashed line in Fig. 2a) in sharp contrast to vanishing photocurrents in pristine regions. This phenomenon is reproducible in other samples (see Supplementary Fig. 7). Scanning photocurrent spectroscopy of total $I_{ph}$ further confirms the observation as shown in Fig. 2c. Moreover, we measured photocurrent along $x$-direction $I_{ph}(x)$ at fixed $y$ position (indicated by the black dashed line in Fig. 2a). As shown in Fig. 2d, consistent and robust valley and peak features are observed at ↑ and ↓ GBs, respectively. The above results show excellent agreement with IPVE theory.

To further demonstrate the effectiveness of IPVE theory, we check if a large polarization-independent photocurrent term $J_y^{Pol-indp}$ exists in ReS2 GBs. Figure 2e shows the polarization-resolved photocurrents in middle of ↑ and ↓ GBs. The polarization-dependent term $J_y^{Pol-dp}$ is complicated since it is influenced by both anisotropic properties of ReS2 domains and IPVE. Hence, we mainly focused on the polarization-independent term $J_y^{Pol-indp}$. Besides, polarization-independent term is more appealing for energy harvesting applications due to the unpolarized nature of sunlight. We further extract $J_y^{Pol-indp}$ and show the mapping results in Fig. 2f. The opposite directions of $J_y^{Pol-indp}$ are observed in ↑ and ↓ GBs, consistent with IPVE theory.

We then investigated the electrical tunability of IPVE in ReS2 GBs using gate bias. Figure 3a shows a device based on few-layer ReS2 with GBs. The thickness of ReS2 flake is 8 nm. The few-layer ReS2 photo-transistor exhibits n-type characteristics (see Supplementary Fig. 8),

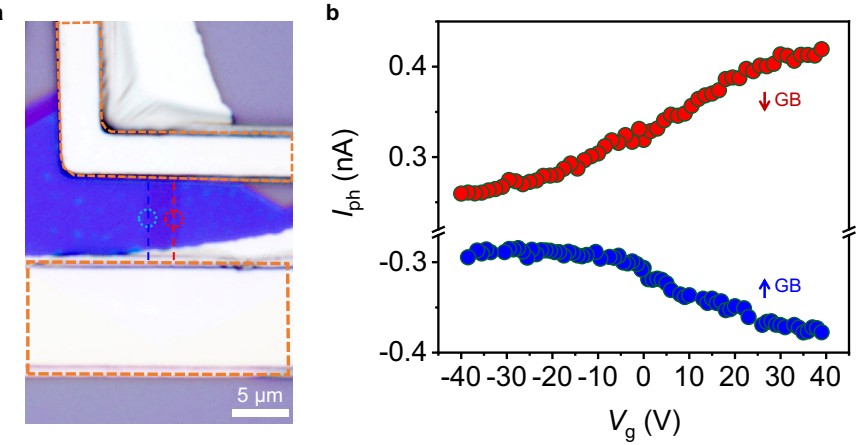

**Fig. 3 | Electrical tunability of IPVE in few-layer ReS2 with GBs. a** Optical image of a phototransistor based on few-layer $ReS_2$ with GBs under polarized white light. Orange dashed lines enclose the electrode areas. Scale bar is 5 μm. **b** Measured

short-circuit photocurrent as a function of gate voltage. The ↑ and ↓ GBs are marked by blue and red dashed lines, respectively. The light spots illuminated at ↑ and ↓ GBs are marked by blue and red circles (presented in **a**), respectively.

consistent with previous reports[38,40]. Opposite directional photocurrents are observed at ↑ and ↓ GBs (see Fig. 3b), showing good agreement with other samples. Moreover, IPVE-induced photocurrents can be effectively tuned by gate voltage with ~ 28% and 60% enhancement from −40 to 40 V for ↑ and ↓ GBs, respectively. To understand this gate tunability of IPVE-induced photocurrent, we first examine whether it simply originates from the tuned Schottky barrier at metal-ReS₂ interface which may affect the collection efficiency of carriers. As shown in Supplementary Fig. 9, the 8 nm-thick ReS₂ device shows a good linear current-voltage characteristic at various gate voltages, indicating a good Ohmic contact between ReS₂ and metal electrodes. Hence, if there exists Schottky barrier, it would be very low which is unlikely to significantly affect the photocurrent intensities. On the other hand, IPVE-induced photocurrents have two contributions which are shift and ballistic currents[6,19,20,41,42]. Shift and ballistic currents strongly depend on the properties of nonequilibrium carriers excited by polarized lights. Thermalization of nonequilibrium carriers can be caused by electron-defect, electron-phonon, and electron-electron interactions. At different gate voltages, electron concentration changes which probably affects the thermalization processes of excited nonequilibrium carriers, such as their mean free bath length and mobility, and hence affects induced IPVE photocurrent densities. This is one plausible explanation. Further studies can be conducted to fully understand this phenomenon.

We compared the strength of IPVE in ReS₂ GBs with other materials. Although structures of ReS₂ GBs are well defined, the effective width of GBs, which denotes the active region with strong inversion symmetry breaking for generating IPVE photocurrent, is unknown. Here, we give a photocurrent density range when effective width varies from 3 to 300 nm. As shown in Fig. 4, the photocurrent densities in ReS₂ GBs are comparable to those in 1D WS₂ nanotube[5], strained 3R-MoS₂[19] and MoS₂[20] and orders of magnitude higher than those in ferroelectric materials[3,4,17,43,44].

## Discussion

To better understand the giant IPVE photocurrent densities in ReS₂ GBs and its underlying physical mechanism, the first-principles calculations of band properties are performed. Detailed information about calculations can be found in the Method Section and Supplementary Information (see Supplementary Figs. 10, 11). As shown in Fig. 5a, ReS₂ near GBs has a lower conduction band minimum and higher valence band maximum compared with those of pristine ReS₂. In addition, GBs have significant influence to the band structures of ReS₂ near GBs through introducing significant number of new states (see

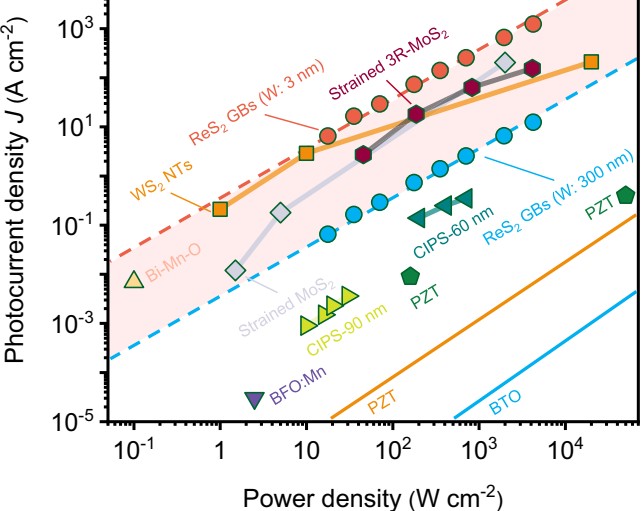

**Fig. 4 | IPVE-induced photocurrent density in various materials.** The orange and blue solid dots indicate IPVE-induced photocurrent density in 8 nm-thick ReS₂ GBs samples. The orange and blue dashed lines are linear fitting curves to the experimental data. The shaded area indicates the photocurrent density range when effective width varies from 3 to 300 nm. Data for other materials are taken from the literature (BaTiO₃ (BTO), ref. 3; Pb(ZrTi)O₃ (PZT), ref. 3,4; Mn-doped BiFeO₃ (BFO:Mn), ref. 43; Bi-Mn-O composite, ref. 44; thin-film CuInP₂S₆ with thickness of 60 and 90 nm (CIPS-60 and −90 nm), ref. 17; strained MoS₂, ref. 20; strained 3R-MoS₂, ref. 19; WS₂ NTs, ref. 5). NTs is short for nanotubes; W is short for effective width.

Supplementary Figs. 10, 11). These new states might improve the light absorption and enhance the IPVE photocurrent. Importantly, a quantum-well structure is formed along x-direction (normal to GB direction) due to lower conduction band minimum and higher valence band maximum near GBs as shown in Fig. 5b. This indicates that carriers generated near GBs tends to be caught into the quantum well and transport along GBs (carrier collection direction of electrodes) is more preferred than other directions. This further enhances the IPVE photocurrent. Besides, the well-defined structures of GBs without any dangling bonds and the indirect bandgap of ReS₂ near GBs could further suppress scatterings and recombination of photo-excited carriers (see Supplementary Fig. 10). These are possible reasons that lead to the giant IPVE photocurrent density in ReS₂ GBs. As shown in Supplementary Fig. 12, we still can observe pronounced IPVE photocurrent at

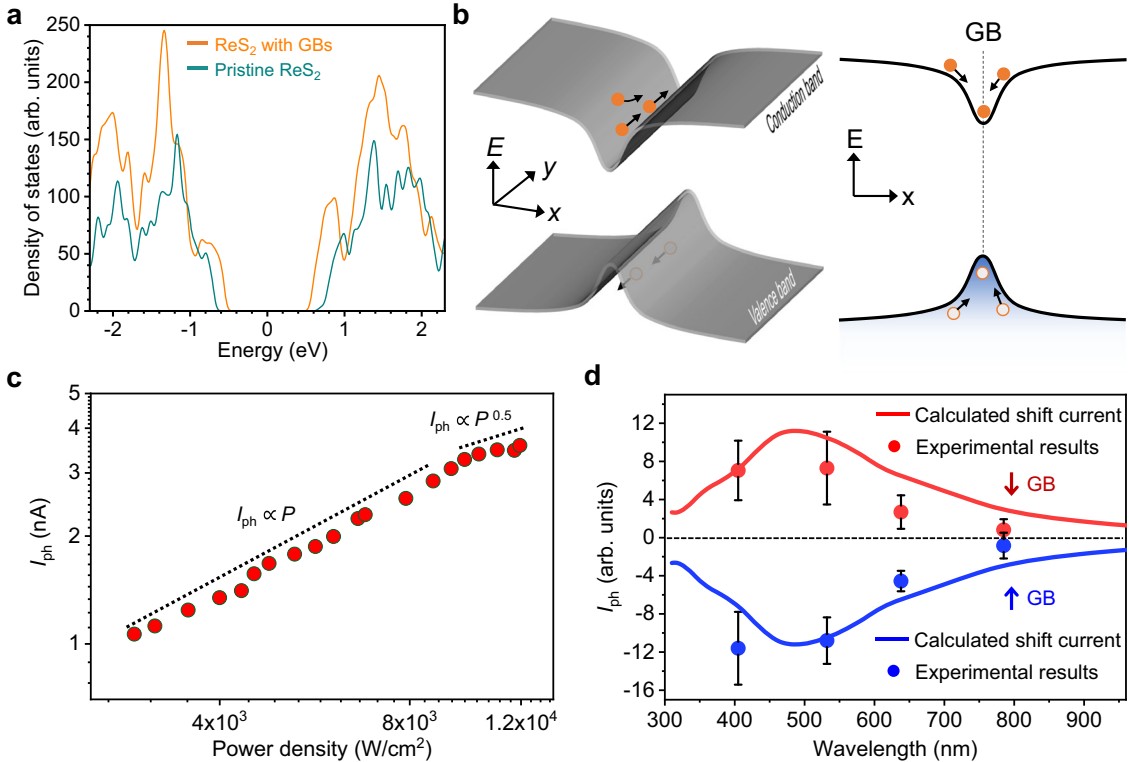

**Fig. 5 | Mechanism of giant IPVE in ReS2 GBs. a** Calculated density of states of $ReS_2$ near GBs and pristine $ReS_2$. **b** Schematic of band diagram of $ReS_2$ near GBs. Quantum well structures are formed along the *x*-direction. The photo-excited electrons and holes are represented by filled and empty circles, respectively. The transport directions of these carriers are marked by black arrows. **c** The power-dependence of IPVE-induced photocurrent of GBs in $ReS_2$ samples. Dashed lines serve as guidelines for linear and square-root dependence. **d** Wavelength-dependent photocurrents at ↑ and ↓ GBs. The calculated and experimental results at ↑ (↓) GBs are represented by blue (red) solid lines and dots, respectively. Error bars denote the standard deviation of the photocurrent measured at GBs.

$ReS_2$ GBs with channel length over 100 μm. In addition, we also studied IPVE at $ReS_2$ edges for comparison, since edges are non-centrosymmetric with broken periodic structures. We fabricated and measured three $ReS_2$ samples in which we did not found detectable IPVE-induced photocurrent at edges (see Supplementary Fig. 13). High density of defect states at edges, such as dangling bonds, might induce strong electron-defect scatterings and suppress the IPVE photocurrent[45].

Power- and wavelength- dependent photocurrents are measured to further clarify the physical mechanism of IPVE observed in $ReS_2$ GBs. As shown in Fig. 5c, the power-dependent photocurrent at GBs shows a transition from linear to square-root dependence when power increases which is consistent with the prediction of theoretical shift current model and previous experimental reports[5,19,20,42,46]. We theoretically calculated the shift current in $ReS_2$ GBs. Detailed calculation process and discussion are shown in Supplementary Information (see Supplementary Note 3 and Supplementary Fig. 14). As shown in Fig. 5d, our shift current model shows good agreement with experimental results at different excitation wavelengths. All above results suggest that shift current dominates the photocurrent generation process of IPVE in $ReS_2$ GBs.

Finally, we conclude through discussing the distinctive aspects of GB-induced symmetry breaking and its potential implications relative to prior research. Firstly, GBs widely exist in all kinds of materials and have a variety of configuration, which provides a capacious platform for IPVE and physics investigations. Secondly, GBs are embedded in bulk materials and there is no symmetry requirement for the crystalline structure of bulk material to induce symmetry breaking in GBs. Thirdly, formation of the quantum-well structure makes GBs a good 1D/quasi-1D system for IPVE investigation which can effectively suppress carrier dissipation to other directions. Fourthly, compared with

edges[45], GBs with well-defined crystalline structures are free of dangling bonds. The reduced electron-defect scatterings in GBs with well-defined structures might suppress scatterings of photo-excited carriers and enhance IPVE photocurrent. Lastly, structures and densities of GBs can be generated and controlled through adjusting material growth conditions[47,48]. Other approaches, such as external strain, can also generate and control GBs in materials[29]. The ability to control formation and structures of GBs is important for making efficient optoelectronic devices. Hence, we believe the rich species and configurations, well-defined 1D/quasi-1D structures, and potential controllability make GBs a promising optoelectronic platform for novel physics and device applications.

## Methods

### Sample preparation
$ReS_2$ samples were prepared on silicon substrate covered with 300 nm $SiO_2$ through standard mechanical exfoliation method. Angle-resolved polarized optical microscope is used to identify GBs and domains in $ReS_2$. Electrodes (5/35 nm Cr/Au) were patterned via standard photolithography process (MicroWriter ML3, Durham Magneto Optics Ltd).

### STEM characterizations
The STEM/HAADF images were obtained using a JEOL ARM200F equipped with a CEOS aberration corrector. The microscope featured a cold field emission gun and was operated at an accelerating voltage of 200 kV. The convergence angle was ~28 mrad.

### Optical characterizations
The devices were characterized using a semiconductor parameter analyzer (FS-Pro) under vacuum (~$10^{-6}$ mbar) at room temperature. For short-circuit photocurrent measurements, 532 nm lasers were used as

excitation sources with laser power of 200 μW, respectively. The lasers were focused by a 50× microscope objective lens (0.5 N.A.). The size of laser spot with a Gaussian profile was ~3 μm for 532 nm laser. Angle-resolved polarized Raman spectroscopy was performed using a 532 nm laser with a spectrometer (Andor SR-500i-D2). A linear polarizer and half-wave plate (Thorlabs) were used to adjust the orientation of the laser polarization.

## Theoretical calculation

Density functional theory (DFT) calculations for structure optimization and electronic properties were performed using the Vinna ab initio simulation package (VASP)[49]. Exchange-correction functional was treated within the generalized gradient approximation of Perdew, Burke, and Ernzerhof[50]. The electronic wave functions were expanded using a planewave basis set with an energy cutoff of 300 eV, and the tolerance for the total energy was $<10^{-4}$ eV. A $1 \times 10 \times 1$ k-mesh was utilized for self-consistent calculations of the supercell structure.

## Data availability

Relevant data supporting the key findings of this study are available within the article and the Supplementary Information file. All raw data generated during the current study are available from the corresponding authors upon request.

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

## Acknowledgements

The work was financially supported by the National Natural Science Foundation of China (62275117, X.C.; 62261136552, J.M.; 52273279, X.Zhao), Shenzhen Excellent Youth Program (RCYX20221008092900001, X.C.), Open research fund of State Key Laboratory of Infrared Physics, Shanghai Institute of Technical Physics, Chinese Academy of Sciences (SITP-NLIST-YB-2022-05, X.C.), Shenzhen Basic Research Program (20220815162316001, X.C.), Natural Science Foundation of Guangdong Province (2023A1515011852, X.L.Y.), Guangdong Major Talent Project (2019QN01C177, X.C.; 2019CX01X014, T.W.), Fundamental Research Funds for the Central Universities (X.Zhao), and Beijing Natural Science Foundation (Z220020 X.Zhao).

## Author contributions

X.C. conceived and supervised the projects. Y.Z. fabricated ReS$_2$ samples and devices with the assistance of Z.L. Y.Z. characterized photocurrent of devices with the assistance of Z.L. and T.W. X.L.Y. did the theoretical calculations. X.Zhou conducted the STEM characterizations with the assistance of X.Zhao, X.C., J.M. and Y.Z. proposed the IPVE/BPVE mechanisms. Y.Z. and X.C. drafted the manuscript with assistance of X.L.Y., X.Zhou and J.M. All authors discussed and commented the manuscript.

## Competing interests

The authors declare no competing interests.
