## [Peer Review File · Nature Communications]

Giant intrinsic photovoltaic effect in one-dimensional van der Waals grain boundariesREVIEWER COMMENTS

Reviewer #1 (Remarks to the Author):

The authors have observed giant bulk photovoltaic effect in grain boundaries region of ReS₂. The experimental data are of high quality. This is a beautiful work: application wise, it may take some time; scientifically, this is a nice model system to study bulk photovoltaic effect. I suggest publishing this manuscript with the following comments.

1. It appears that the point group of the A/B or B/A grain boundary is C_{2v} (two mirror planes and one C₂ rotation axis). Can the authors confirm this?
2. There is a typo here: "at the bottom conductance and top valence bands"
3. "ReS₂ GBs has a smaller bandgap compared with pristine ReS₂". How do you define band gap for grain boundary? Do these grain boundaries contribute to the mini-states like
4. I am not convinced by the results in Figure 5 and the relevant narratives. I suggest simply removing it. If the authors want to keep it, please conduct calculations on photo current densities. Removing this part would not hurt the paper at all (the experimental results are great already).
5. What is the mechanism for gate's tunability?

Reviewer #2 (Remarks to the Author):

Y. Zhou et al. present the development of a substantial BPVE at the grain boundary of ReS₂. The presence of the grain boundary is convincingly demonstrated through STEM and angle-resolved Raman data. The short-circuit photocurrent measurements provide direct evidence of the BPVE at the boundary. All experimental data appear consistent and reliable, supporting the potential of ReS₂ grain boundaries as a novel platform for realizing a large photovoltaic effect.

Despite the successful observation of the BPVE, the current version cannot be recommended for publication in Nature Communications for the following reasons:

1. The generation of photocurrent in transition metal dichalcogenides (TMDs) is known to result from symmetry breaking, a well-established concept. While the grain boundary approach offers a fresh perspective on breaking symmetry, the manuscript does not sufficiently highlight the novelty and significance of this approach in the scientific context. It would be beneficial to emphasize the distinctive aspects of grain boundary-induced symmetry breaking and its potential implications relative to prior research.
2. The BPVE is observed using a single photon source; however, the manuscript lacks an exploration of the relationship between photocurrent and light frequency or intensity. Furthermore, the manuscript does not investigate the distinction between the true edge and the grain boundary. The absence of an analysis of light dependence and position dependence on photocurrent generation hinders a

comprehensive understanding of this intriguing BPVE developed at the grain boundary.

3. The manuscript lacks a discussion of the microscopic mechanism underlying the giant BPVE. There is no evidence that the band edge transition is responsible for the large BPVE. While the reduced recombination process can be an important factor, it does not directly create the photocurrent. Thus, a plausible explanation is required to understand the observed large photocurrent.

To enhance the manuscript's quality and address these concerns, further experiments, discussions, and contextualization of results are needed before considering it for publication in Nature Communications.

Reviewer #3 (Remarks to the Author):

Intrinsic photovoltaic effect reflecting the symmetry breaking of solids, is now attracting much attention due to the potential of overcoming the Shockley-Queisser limit in the conventional solar cells made of semiconducting p-n junctions and also its mechanism related with the carrier dynamics or band geometry/topology. In this paper, authors reported the giant bulk photovoltaic effect in one-dimensional ReS₂ grain boundary and successfully demonstrated that observed giant photocurrent response precisely reflect the symmetry breaking at the grain boundaries. I think these findings are very interesting, providing a new design principle of symmetry engineering in two-dimensional materials and resultant functional devices.

I have several comments and questions below.

1. Do authors have any ideas about the origins of the observed intrinsic photovoltaic effect? For example, shift current, which is one mechanism of the photovoltaic effect induced by the linearly polarized light, may show the characteristic wavelength dependence. Can authors measure it? Also, can authors calculate the shift current based on their calculated band structure?
2. Related with the above comments, I would like to know the authors' opinion on why photocurrent is enhanced by applying the positive gate voltage.
3. I am interested in whether we can control (or intentionally create) the grain boundary or not. Controllability of the grain boundary might be important for making the efficient photovoltaic devices.
4. I think edge of the sample also have a similar symmetry breaking as the grain boundary. Does photocurrent appear at the edge of the sample?
5. I am wondering that the word of "intrinsic photovoltaic effect" might be better than "bulk photovoltaic effect" in the present case because the grain boundary is not the bulk of the sample.

Point-by-point response to referee 1

The authors have observed giant bulk photovoltaic effect in grain boundaries region of ReS₂. The experimental data are of high quality. This is a beautiful work: application wise, it may take some time; scientifically, this is a nice model system to study bulk photovoltaic effect. I suggest publishing this manuscript with the following comments.

Response: We thank reviewer for his/her positive comments.

1. It appears that the point group of the A/B or B/A grain boundary is C_{2v} (two mirror planes and one C₂ rotation axis). Can the authors confirm this?

Response: We are very grateful to reviewer for his/her valuable comments. After closely examining the atomic configuration at the grain boundary (GB) with high-resolution STEM (Figure R1a) and comparing it with the DFT-optimized atomic model, we have concluded that the only symmetry present at GB is the two-fold rotation axis along the y-direction, as shown in Figure R1b.

We also evaluated other possible symmetry elements at GBs. Figure R1c illustrates a

potential two-fold rotational axis along the z -direction, which is found to be absent due to the non-equivalent distances from this axis to neighboring Re atoms (4.49 Å vs. 4.77 Å as indicated in Fig. R1c). Additionally, the angle formed by Re atoms around this axis is 176.8°, not fulfilling the criteria for a C_2 rotational axis. Our examination of a mirror plane along the xz plane, as presented in Figure R1d, shows unequal distances from two neighboring Re atoms to this plane (3.56 Å vs. 3.01 Å). Finally, we considered a potential mirror plane along the yz plane. While it appears to exist from a top view, side view analysis clearly shows that such a mirror plane does not exist, especially in the arrangement of S atoms. In summary, we confirm that GB possesses only a two-fold rotational axis, and the point group for the local structure around the boundary should be classified as C_2 .

In the revised manuscript, we have clearly pointed out the point group it belongs (see Page 5 in the main text). We also incorporated the above discussion in the Supplementary Information (see Supplementary Note 1 and Supplementary Fig. 3).

Figure R1. Symmetry analysis at GB. **a**, Atomic-resolution STEM image of the boundary overlaid with the atomic model, where blue dots represent Re atoms and pink

dots indicate S atoms. Scale bar: 5 Å. **b**, Detailed atomic model of the grain boundary highlighting the two-fold rotational axis along with the y -direction. **c-e**, Examination of other potential symmetry elements at the boundary: a possible two-fold rotational axis around the z -direction (c), a potential mirror plane parallel to the xz plane (d), and a possible mirror plane parallel to the yz plane (e). All these symmetry does not exist.

2. There is a typo here: “at the bottom conductance and top valence bands”

Response: Thanks for pointing out this typo. In the revised manuscript, we revise the expression into: “at valence band maximum and conductance band minimum”.

3. “ReS₂ GBs has a smaller bandgap compared with pristine ReS₂”. How do you define band gap for grain boundary? Do these grain boundaries contribute to the mini-states like?

Response: We thank reviewer for pointing out this issue. We apologize that the original sentence “ReS₂ GBs has a smaller bandgap compared with pristine ReS₂” is ambiguous and not accurate. The system we investigated is “ReS₂ with GBs” or “ReS₂ near GBs”. DFT calculated band structures are referred to “ReS₂ with GBs” system not “ReS₂ GBs”. The bandgap can be referred to the bandgap of ReS₂ with GBs. As shown in Figure R2 below, we further compared calculated band structures of pristine ReS₂ and “ReS₂ with GBs” system. GBs have significant influence to the band structures of ReS₂ near GBs through introducing significant number of new states.

In the revised manuscript, the sentence “ReS₂ GBs has a smaller bandgap compared with pristine ReS₂” has been changed into: “ReS₂ near GBs has a lower conduction band minimum and higher valence band maximum compared with those of pristine ReS₂. In addition, GBs have significant influence to the band structures of ReS₂ near GBs through introducing significant number of new states” (see Page 10 in the main text). Figure R2 has been included in Supplementary Information (see Supplementary Fig. 10).

Figure R2. Comparison of band structures of pristine ReS₂ and ReS₂ with GBs.

4. I am not convinced by the results in Figure 5 and the relevant narratives. I suggest simply removing it. If the authors want to keep it, please conduct calculations on photo current densities. Removing this part would not hurt the paper at all (the experimental results are great already).

Response: Thank you for the suggestion. In the revised manuscript, we have removed previous Figure 5. In addition, we have added more experimental results and calculations on the mechanism of the giant BPVE near ReS₂ GBs in the new Figure 5 (see Page 11 in the main text).

Figure R3. The new Figure 5. **a**, Calculated density of states of ReS₂ GBs and pristine ReS₂. **b**, Schematic of band diagram of ReS₂ near GBs. Quantum well structures are formed along *x*-direction. **c**, The power-dependence of BPVE-induced photocurrent of GBs in ReS₂ samples. Dashed lines serve as guidelines for linear and square-root dependence. **d**, Wavelength-dependent photocurrent at ↑ and ↓ GBs.

In addition, we have theoretically calculated the shift current densities in ReS₂ with grain boundaries. As shown in Fig. R3d, calculated wavelength-dependent shift currents show good agreement with experimental data. Besides, power-dependent photocurrent shows a transition from $I_{\text{ph}} \propto P$ to $I_{\text{ph}} \propto P^{0.5}$ when power increases which is consistent with the shift current model as shown in Fig. R3c (Y. J. Zhang et al., Nature 570, 349 2019; Jiang, J. et al. Nat. Nanotechnol. 16, 894 ,2021; Dong, Y. et al. Nat. Nanotechnol. 18, 36–41, 2023; Tan, L. Z. et al. npj Comput. Mater. 2, 16026, 2016; Morimoto, T. et al. Sci. Adv. 2, e1501524, 2016). Detailed calculation process of shift current has been shown in Supplementary Note 3 and Supplementary Fig. 14.

5. What is the mechanism for gate's tunability?

Response: We thank the referee for his/her valuable comments. We first examine whether the tunability simply originates from the tuned Schottky barrier at metal-ReS₂ interface which may affect the collection efficiency of carriers. As shown in Fig. R4, the 8 nm-thick ReS₂ device shows a good linear current-voltage characteristic, indicating a good Ohmic contact between ReS₂ and metal electrodes. Hence, if there exists Schottky barrier, it would be low which is unlikely to affect the photocurrent values significantly.

On the other hand, BPVE-induced photocurrents have two contributions which are shift and ballistic currents (Burger, A. M. et al. Sci. Adv. 5, eaau5588, 2019; Cook, A. M. et al. Nat. Commun. 8, 14176, 2017; Dong, Y. et al. Nat. Nanotechnol. 18, 36–41, 2022; Jiang, J. et al. Nat. Nanotechnol. 16, 894-901, 2021). Shift and ballistic currents strongly depend on the properties of nonequilibrium carriers excited by polarized lasers. Thermalization of nonequilibrium carriers can be caused by electron-defect, electron-

phonon, and electron-electron interactions. At different gate voltages, electron concentration changes which probably affect the thermalization process of excited nonequilibrium carriers, such as their mean free bath length and mobility, and hence affect the induced BPVE photocurrent. This is one plausible explanation. Further studies can be conducted to fully understand this phenomenon.

In the revised manuscript, we have included above discussion in the main text (see Page 9). Fig. R4 has been included in Supplementary Information (see Supplementary Fig. 9).

Figure R4. I_{ds} – V_{ds} curves of the 8 nm ReS₂ device shows linear characteristic with gate voltage from -40 V to 40V.

Point-by-point response to referee 2

Y. Zhou et al. present the development of a substantial BPVE at the grain boundary of ReS₂. The presence of the grain boundary is convincingly demonstrated through STEM and angle-resolved Raman data. The short-circuit photocurrent measurements provide direct evidence of the BPVE at the boundary. All experimental data appear consistent and reliable, supporting the potential of ReS₂ grain boundaries as a novel platform for realizing a large photovoltaic effect.

Despite the successful observation of the BPVE, the current version cannot be recommended for publication in Nature Communications for the following reasons:

Response: We thank reviewer for his/her positive assessment and insightful comments.

1. The generation of photocurrent in transition metal dichalcogenides (TMDs) is known to result from symmetry breaking, a well-established concept. While the grain boundary approach offers a fresh perspective on breaking symmetry, the manuscript does not sufficiently highlight the novelty and significance of this approach in the scientific context. It would be beneficial to emphasize the distinctive aspects of grain boundary-induced symmetry breaking and its potential implications relative to prior research.

Response: Thank you for the valuable suggestions. In the revised manuscript (see Page 12 in the main text), we have further emphasized the distinct aspects of grain boundary-induced BPVE and its potential implications in the conclusion section. For your convenience, we put the revised text below:

“Finally, we conclude through discussing the distinctive aspects of GB-induced symmetry breaking and its potential implications relative to prior research. Firstly, GBs widely exist in all kinds of materials and have a variety of configuration, which provides a capacious platform for IPVE and physics investigations. Secondly, GBs are embedded in bulk materials and there is no symmetry requirement for the crystalline structure of bulk material to induce symmetry breaking in GBs. Thirdly, formation of the quantum-well structure makes GBs a good 1D/quasi-1D system for IPVE investigation which can effectively suppress carrier dissipation to other directions. Fourthly, compared with edges, GBs with well-defined crystalline structures are free of dangling bonds. The reduced electron-defect scatterings in GBs with well-defined structures might suppress scatterings of photo-excited carriers and enhance IPVE photocurrent. Lastly, structures

and densities of GBs can be generated and controlled through adjusting material growth conditions. Other approaches, such as external strain, can also generate and control GBs in materials. Hence, we believe the rich species and configurations, well-defined 1D/quasi-1D structures, and potential controllability make GBs a promising optoelectronic platform for novel physics and device applications.”

2. The BPVE is observed using a single photon source; however, the manuscript lacks an exploration of the relationship between photocurrent and light frequency or intensity. Furthermore, the manuscript does not investigate the distinction between the true edge and the grain boundary. The absence of an analysis of light dependence and position dependence on photocurrent generation hinders a comprehensive understanding of this intriguing BPVE developed at the grain boundary.

Response: We thank reviewer for his/her insightful suggestions. Since the previous sample has been sent for STEM characterization, we have prepared a new sample to further investigate wavelength- and position- dependent BPVE photocurrents (see Fig. R5 below).

Firstly, polarized 405, 532, 638 and 785 nm lasers with the same incident power ($28.2 \mu\text{W}/\mu\text{m}^2$) were focused on the grain boundaries and short-circuit photocurrents (I_{ph}) were collected (the angle between laser polarization and x -axis is $\sim 30^\circ$). As shown in Fig. R5b, the BPVE-induced photocurrents show clear wavelength-dependence. Then, we theoretically calculated shift currents along ReS_2 GBs. Detailed calculation processes are shown in Supplementary Note 3 and Supplementary Fig. 14. As shown in Fig. R6, theoretically calculated wavelength-dependent shift currents show good agreement with experimental data. This indicates that shift current model can explain the observed BPVE at ReS_2 GBs quite well.

Figure R5. **a**, The polarized optical image of the sample with grain boundary. The \uparrow and \downarrow GBs are marked by blue and red dash line, respectively. **b**, Wavelength-dependent I_{ph} of the sample. The laser was used to excited at the middle of the channel. **c-f**, I_{ph} measured with incident 405, 532, 638 and 785 nm lasers along \uparrow and \downarrow GBs. Electrodes are marked by yellow regions. The power of all lasers was 200 μ W.

Figure R6. Theoretically calculated shift currents and experimental results.

Secondly, we have investigated the power-dependence of photocurrent at GBs. As shown in Fig. R7 below, the power-dependent photocurrent at GBs shows a transition from $I_{ph} \propto P$ to $I_{ph} \propto P^{0.5}$ when power increases which is consistent with the shift current model (Y. J. Zhang et al., Nature 570, 349 2019; Jiang, J. et al. Nat. Nanotechnol. 16, 894 ,2021; Dong, Y. et al. Nat. Nanotechnol. 18, 36–41, 2023; Tan, L. Z. et al. npj Comput. Mater. 2, 16026, 2016; Morimoto, T. et al. Sci. Adv. 2, e1501524, 2016).

Figure R7. The power-dependence of BPVE-induced photocurrent of GBs in ReS₂ samples. Dashed lines serve as guidelines for linear and square-root dependence.

Thirdly, we fabricated more samples without GBs to investigate photocurrents at true edges. A linearly polarized 532 nm laser with incident power density of (28.2 $\mu\text{W}/\mu\text{m}^2$) is used. As shown in Fig. R8, I_{ph} measured along the edges show ordinary shapes with vanishing value in the middle of channel. This indicates that although edges with broken symmetry theoretically support BPVE, it might be too small to be detected.

Figure R8. Characterizations of edge photocurrents in ReS₂ samples without GBs. I_{ph} measured along the edges as indicated by the solid lines. Electrodes are marked by green shadows. The photocurrent vanished in the middle of ReS₂ channels.

In summary, wavelength-, position-, and power-dependent photocurrents and theoretical calculations have been performed to better understand the mechanism of

BPVE developed at grain boundaries. Shift current model shows a good agreement with our experimental observations.

In the revised manuscript, Fig. R6 and R7 have been included in the main text (see Fig. 5). Fig. R5a, R5c-f, and R8 have been included in the Supplementary Information (see Supplementary Fig.12 and 13). The above discussions have been included in the revised manuscript (see Page 10-12 in the main text) and Supplementary Information (see Supplementary Note 3).

3. The manuscript lacks a discussion of the microscopic mechanism underlying the giant BPVE. There is no evidence that the band edge transition is responsible for the large BPVE. While the reduced recombination process can be an important factor, it does not directly create the photocurrent. Thus, a plausible explanation is required to understand the observed large photocurrent.

Response: This is an insightful comment. To explain the large BPVE photocurrent near grain boundary (GB), we further explore the DFT calculation results and band properties of ReS₂ near GBs. Fig. R9 compares the band structures of pristine ReS₂ and ReS₂ near GBs. A large number of new bands and states are generated around GBs. More than half of 14 conduction bands and 14 valence bands near Fermi level contribute to large number of electron states around GBs (see Fig. R10). These probably enhance the optical absorption of ReS₂ near GBs and hence enlarge the numbers of nonthermalized carriers under light excitation.

Figure R9. The band structures of pristine ReS₂ (left) and ReS₂ near GBs (right).

Figure R10. The band properties and distribution of electron states of ReS₂ near GBs. **a, b**, Band structures of ReS₂ near GBs. 14 conduction bands (**a**) and 14 valence bands (**b**) near Fermi level were analyzed. **c, d**, Distribution of electron states contributed by conduction bands (**a**) and valence bands (**b**) near Fermi level. The GBs are marked by blue regions. Large number of electron states are found around GBs.

On the other hand, quantum well structures along x -direction (normal to GB direction) are formed near GBs as shown in Fig. R11. This indicates that carriers generated near GBs tends to be caught into the quantum well and travel along GBs (y -direction) instead of dissipating to other directions. This could further enhance the BPVE photocurrent. Besides, the well-defined structures of GBs without any dangling bonds and the indirect bandgap of ReS₂ near GBs could further suppress scatterings and recombination of photo-excited carriers. These are possible reasons that lead to the giant BPVE photocurrent density in ReS₂ GBs.

In the revised manuscript, we have included above discussions in the main text (see Page 10-11). In addition, Fig. R11 has been included in the main text (see Fig.5) and Fig. R9 and R10 have been included in Supplementary Information (see Supplementary Fig. 10 and 11).

Figure R11. Formation of quantum well structures near GBs. **a**, Density of states of ReS₂ with GBs and pristine ReS₂. **b**, Schematic of quantum well structures near GBs. The black dash line indicates the position of GB. The carriers generated near GBs are trapped by the quantum well structure and travel along GBs (y-direction) instead of dissipating to other directions.

To enhance the manuscript's quality and address these concerns, further experiments, discussions, and contextualization of results are needed before considering it for publication in Nature Communications.

Response: We would like to thank you for your thoughtful comments and suggestions. We truly appreciate the time and efforts you invested in our paper. With your help, our paper has been improved substantially. We believe the manuscript has been improved significantly and meets the high standard of Nature Communications. We summarize the main improvement below:

Firstly, we have demonstrated that shift current model can describe the observed IPVE quite well through analyzing the power- and wavelength- dependent photocurrent at GBs.

Secondly, we have theoretically calculated the shift current in ReS₂ GBs based on the first-principles calculations and tight-binding model, which shows good agreement with experimental results.

Thirdly, the reasons for observation of giant photocurrent BPVE at ReS₂ GBs are explained in details. Especially, the formation of quantum-well structures near GBs is essential for the giant photocurrent.

Fourthly, original Figure 5 has been removed. The new Figure 5 mainly focus on understanding the underlying physical mechanism of BPVE in ReS₂ GBs.

Finally, we have made every effort to fully address all of your other concerns and suggestions.

Point-by-point response to referee 3

Intrinsic photovoltaic effect reflecting the symmetry breaking of solids, is now attracting much attention due to the potential of overcoming the Shockley-Queisser limit in the conventional solar cells made of semiconducting p-n junctions and also its mechanism related with the carrier dynamics or band geometry/topology. In this paper, authors reported the giant bulk photovoltaic effect in one-dimensional ReS₂ grain boundary and successfully demonstrated that observed giant photocurrent response precisely reflect the symmetry breaking at the grain boundaries. I think these findings are very interesting, providing a new design principle of symmetry engineering in two-dimensional materials and resultant functional devices.

I have several comments and questions below.

Response: We thank reviewer for his/her positive assessment of our manuscript.

1. Do authors have any ideas about the origins of the observed intrinsic photovoltaic effect? For example, shift current, which is one mechanism of the photovoltaic effect induced by the linearly polarized light, may show the characteristic wavelength dependence. Can authors measure it? Also, can authors calculate the shift current based on their calculated band structure?

Response: We thank reviewer for his/her insightful comments. Intrinsic photovoltaic effect (IPVE)-induced photocurrents have two contributions which are shift current and ballistic current. We further performed power- and wavelength- dependent measurements and theoretical calculations of shift current to further clarify the physical mechanism of IPVE in ReS₂ GBs.

Power-dependent measurements could provide useful information about the mechanism. According to previous theoretical and experimental results (Y. J. Zhang et al., Nature 570, 349 2019; Jiang, J. et al. Nat. Nanotechnol. 16, 894 ,2021; Dong, Y. et al. Nat. Nanotechnol. 18, 36–41, 2023; Tan, L. Z. et al. npj Comput. Mater. 2, 16026, 2016; Morimoto, T. et al. Sci. Adv. 2, e1501524, 2016), shift current would show a power-dependent transition from $I_{\text{ph}} \propto P$ to $I_{\text{ph}} \propto P^{0.5}$ when power increases. As shown in Fig. R12, the power-dependent photocurrent at GBs shows a transition from linear to square-root dependence. This is consistent with previous studies on shift current models. This indicates that the contribution of shift current might be more significant than ballistic current.

Figure R12. Power-dependence of IPVE-induced photocurrent at ReS₂ GBs. Dashed lines serve as guidelines for linear and square-root dependence.

Polarized 405, 532, 638 and 785 nm lasers with the same incident power ($28.2 \mu\text{W}/\mu\text{m}^2$) were focused on the grain boundaries and short-circuit photocurrents (I_{ph}) were collected (the angle between laser polarization and x -axis is $\sim 30^\circ$). As shown in Fig. R13b, the IPVE-induced photocurrents show clear wavelength-dependence. Then, we theoretically calculated shift currents along ReS₂ GBs. Detailed calculation processes are shown in Supplementary Note 3 and Supplementary Fig.14. As shown in Fig. R14, theoretically calculated wavelength-dependent shift currents show good agreement with experimental data. This indicates that shift current model can explain the observed IPVE at ReS₂ GBs quite well.

Figure R13. a, The polarized optical image of the sample with grain boundary. The \uparrow and \downarrow GBs are marked by blue and red dash line, respectively. b, Wavelength-dependent I_{ph} of the sample. The laser was used to excited at the middle of the channel. c-f, I_{ph} measured with incident 405, 532, 638 and 785 nm lasers along \uparrow and \downarrow GBs. Electrodes are marked by yellow regions. The power of all lasers was 200 μ W.

Figure R14. Theoretically calculated shift currents and experimental results.

In summary, wavelength- and power-dependent photocurrents and theoretical calculations have been performed to better understand the mechanism of IPVE developed at grain boundaries. Shift current model shows a good agreement with our experimental observations.

In the revised manuscript, Fig. R12 and R14 have been included in the main text (see

Fig. 5). Fig. R13a and R13c-f have been included in the Supplementary Information (see Supplementary Fig. 12). The above discussions have been included in the revised manuscript (see Page 10-12 in the main text) and Supplementary Information (see Supplementary Note 3).

2. Related with the above comments, I would like to know the authors' opinion on why photocurrent is enhanced by applying the positive gate voltage.

Response: We thank the referee for his/her valuable comments. We first examine whether the tunability simply originates from the tuned Schottky barrier at metal-ReS₂ interface which may affect the collection efficiency of carriers. As shown in Fig. R15, the 8 nm-thick ReS₂ device shows a good linear current-voltage characteristic, indicating a good Ohmic contact between ReS₂ and metal electrodes. Hence, if there exists Schottky barrier, it would be low which is unlikely to affect the photocurrent values significantly.

On the other hand, IPVE-induced photocurrents have two contributions which are shift and ballistic currents (Burger, A. M. et al. *Sci. Adv.* 5, eaau5588, 2019; Cook, A. M. et al. *Nat. Commun.* 8, 14176, 2017; Dong, Y. et al. *Nat. Nanotechnol.* 18, 36–41, 2022; Jiang, J. et al. *Nat. Nanotechnol.* 16, 894-901, 2021). Shift and ballistic currents strongly depend on the properties of nonequilibrium carriers excited by polarized lasers. Thermalization of nonequilibrium carriers can be caused by electron-defect, electron-phonon, and electron-electron interactions. At different gate voltages, electron concentration changes which probably affect the thermalization process of excited nonequilibrium carriers, such as their mean free bath length and mobility, and hence affect the induced IPVE photocurrent. This is one plausible explanation. Further studies can be conducted to fully understand this phenomenon.

In the revised manuscript, we have included above discussion in the main text (see Page 9). Fig. R15 has been included in Supplementary Information (see Supplementary Fig. 9).

Figure R15. I_{ds} – V_{ds} curves of the 8 nm ReS₂ device shows linear characteristic with gate voltage from -40 V to 40V.

3. I am interested in whether we can control (or intentionally create) the grain boundary or not. Controllability of the grain boundary might be important for making the efficient photovoltaic devices.

Response: The controllability of the grain boundary is a brilliant idea in realizing different application of the intrinsic photovoltaic effect. Here, we offer some discussions on this issue. Firstly, the grain boundaries can be controlled through changing the growth conditions of materials. For examples, recent researches have demonstrated that grain boundary structures and densities in graphene, hBN, and transition metal dichalcogenides (TMDs) can be effectively controlled through adjusting growth time, temperature and orientation of substrates (S. J. Yang et al., Adv. Mater. 35, e2203425, 2023; He, Y et al., Nat. Commun. 11, 57, 2020). As for ReS₂, direct control of grain boundaries in growth step is still scare, and further studies are encouraged. Alternatively, a recent report suggests that the grain boundaries in ReS₂ can be reconstructed by applying uniaxial strain along *b*-axis of ReS₂ (J. Jeong et al., Adv. Mater. 34, e2108777, 2022), offering another promising method for controllability of grain boundary. Finally, the concept of symmetry engineering in van der waals interfaces might provide new perspective to realize controllability of artificial grain boundaries (Takatoshi Akamatsu et al., Science 372, 68-72, 2021). A recent study shows that the overlapping grain boundaries can be artificially formed by parallelly stacking

two grain edges oriented along the b -axis of ReS₂ (S. Wang et al., Matter 3, 2108-2123, 2020). It would be of interest to see if this artificial grain boundaries will show intrinsic photovoltaic effective or not, and whether can this technique be used to create type-II grain boundaries as presented in our samples.

In the revised manuscript, we have included above discussion in the main text (see Page 12).

4. I think edge of the sample also have a similar symmetry breaking as the grain boundary. Does photocurrent appear at the edge of the sample?

Response: Thanks for the comment. Edges are regions with broken symmetry which theoretically host IPVE. However, in many materials, IPVE-induced photocurrents at edges are too small to be detected. According to reviewer's suggestion, we have fabricated more ReS₂ samples without GBs and performed photocurrent measurements on pure edges of ReS₂. A linearly polarized 532 nm laser with incident power density of 28.2 $\mu\text{W}/\mu\text{m}^2$ were focused on the edges. As shown in Fig. R16, I_{ph} measured along two edges show ordinary shape with vanishing value in the middle of channel in all three samples. Although we didn't observe the IPVE at the edge of ReS₂, a recent study has demonstrated that the strong symmetry breaking and low dimensionality of edges in Weyl semimetal WTe₂ can generate strong BPVE-induced photocurrents (Wang, Q. et al. Nat. Commun. 10, 5736, 2019). This finding indicated that symmetry breaking in edges of materials can be an interesting and important approach for realizing intrinsic photovoltaic effect.

Fig. R16 has been included in the Supplementary Information (see Supplementary Figure 12). Discussions have been included in the revised manuscript (see Page 11 in the main text).

Figure R16. Characterizations of edge photocurrent in three ReS₂ samples. I_{ph} measured along the edges as indicated by the solid lines. The polarization of 532 nm laser was along the edges. Electrodes are marked by green regions. Photocurrent vanish in the middle of ReS₂ channels.

5. I am wondering that the word of “intrinsic photovoltaic effect” might be better than “bulk photovoltaic effect” in the present case because the grain boundary is not the bulk of the sample.

Response: Thanks for the suggestion. We agree with reviewer that intrinsic photovoltaic effect is more appropriate to describe this phenomenon in one-dimensional grain boundaries. We have made relative changes in the title and revised manuscript.

REVIEWERS' COMMENTS

Reviewer #1 (Remarks to the Author):

I am happy with the response to my questions. I suggest publishing it.

Reviewer #2 (Remarks to the Author):

The authors have conducted additional experiments to demonstrate the microscopic origin of the giant photocurrent generation and their explanation appears plausible and sound. Therefore, I recommend the publication of this study in nature communications.

Reviewer #3 (Remarks to the Author):

Authors improved their discussion on the origins/mechanisms of the observed interesting photovoltaic response and revised the manuscript appropriately. I am now satisfied with their response.